# Anti-Quantum Lattice-Based Ring Signature Scheme and Applications in VANETs

**DOI:** 10.3390/e23101364

**Published:** 2021-10-19

**Authors:** Chunhong Jiao, Xinyin Xiang

**Affiliations:** 1School of Physics, Xi’an Jiaotong University City College, Xi’an 710018, China; 2School of Information, Xi’an University of Finance and Economics, Xi’an 710100, China; xxy@xaufe.edu.cn; 3China (Xi’an) Institute for Silk Road Research, Xi’an University of Finance and Economics, Xi’an 710100, China

**Keywords:** anti-quantum, ring signature, lattice-based cryptography, anonymity

## Abstract

Message authentication is crucial because it encourages participants to accept countermeasures and further transmit messages to legitimate users in a network while maintaining the legitimacy of the identity of network members. An unauthorized user cannot transmit false messages to a given network. Although traditional public key cryptography is suitable for message authentication, it is also easy to manage and generate keys, and, with the expansion of an entire network, the system needs a lot of computing power, which creates additional risks to network security. A more effective method, such as ring signature, can realize this function and guarantee more security. In this paper, we propose an anti-quantum ring signature scheme based on lattice, functionality analysis, and performance evaluation to demonstrate that this scheme supports unconditional anonymity and unforgeability. After efficiency analysis, our scheme proved more effective than the existing ring signature schemes in processing signature generation and verification. The proposed scheme was applied to VANETs that support strong security and unconditional anonymity to vehicles.

## 1. Introduction

User privacy protection is one of the main goals of modern cryptography, but a digital signature, as a cryptographic primitive to realize the main functions, such as identity authentication, does not consider privacy as a security goal. The public key of the signer is the necessary information to verify the validity of the signature, so the identity of the signer of the digital signature is always visible to the verifier. This explicit validity verification method cannot meet the needs of users in some scenarios. Cryptography primitives, such as group signature [1] and ring signature [2], focus on the protection of user privacy in the above scenarios. They allow the signer to sign in the name of the group, and the verifier can only confirm that the signature is generated by a user in the group but cannot know the specific identity of the signer. Between the two, the group signature system has the role of group administrator, responsible for managing group members and tracking the identity of signers. The group in the ring signature system is completely self-organized; there is no special organization, and the anonymity of the signature cannot be revoked, which provides a higher level of privacy protection. The premise of the ring signature is similar to that of the group signature, both of which hide the identity of the signer within a certain group, but there are significant differences. In the group signature scheme, the group administrator can revoke the anonymity of the group signature, while, in the ring signature, there is no centralized organization, and the group that hides the identity of the signer can be selected by the signer himself immediately. There is no need for any co-operation among users. This means that the ring signature supports stronger anonymity.

In recent years, message authentication in the blockchain has become extremely vital as it encourages users to accept messages and transmit them to other users in the network. To a certain extent, blockchain [3] technology enables the co-operation and value flow between individuals who do not trust each other. However, the data transmission and storage on blockchain are publicly visible, which can be provided to any information query, and can only protect the privacy of both parties through the form of “pseudo anonymity.” In order to meet the needs of this technology, the ring signature is more likely to solve the privacy protection problem of blockchain so as to meet the need for user identity anonymity and transaction information unforgeability. In contrast to classical cryptography, recent studies have shown that lattices are enjoying widespread interest in cryptography. Lattice-based cryptography is widely believed to be resistant against quantum computers, which prompts us to design secure cryptographic schemes as an ideal candidate. In 2008, Gentry et al. [4] employed a novel technique called preimage sampling function (PSF) and built a lattice-based signature scheme in the random oracle model. In 2009, Buchmann et al. [5] designed a Merkle tree signature scheme in the random oracle model under the worst-case hard lattice problems. Subsequently, Boyen et al. [6] proposed the construction of a short signature from hard lattices without random oracles. Previous lattice-based signature schemes used the trapdoor function on the lattice to generate credentials for users in the group. Because of the system parameters required by the trapdoor, the actual size of the signature was too large. More precisely, to improve the efficiency of the signature, it is natural to ask whether we can design a ring signature scheme with enhanced security and better efficiency, so it seems feasible to construct the cryptosystem as we do from lattices.

### 1.1. Related Works

Rivest et al. [2] first proposed the notion of a ring signature scheme in 2001. They presented a ring signature scheme based on the Rabin trapdoor function and RSA trapdoor permutation and proved the security of the proposed scheme under the random oracle model. In the ring signature, any user can sign any message on behalf of the whole ring, and any verifier who obtains the ring public key can verify whether the signature comes from the ring. It is worth noting that, if only the ordinary ring signature scheme is used to solve the problem of privacy protection in the blockchain, the fund holder can sign the same fund many times under the protection of the ring signature and cannot be detected. Brakerski et al. [7] proposed an efficient general framework to construct ring signature schemes under the standard model. Specifically, the scheme defines the concept of the ring trapdoor function and shows how to construct ring signatures using the ring trapdoor. In 2014, Liu et al. [8] proposed a linkable ring signature with unconditional anonymity; the formal security model definition and proof are given. However, the above ring signature is mainly based on a public key certificate, which has the burden of key management and cannot often avoid the complex problem of the user’s public key certificate management.

Duan et al. [9] presented a ring confidential transaction protocol for blockchain-enabled systems. Because of the threat from quantum computing technology, the traditional cryptosystem based on number theory problems (such as the large integer factorization problem and the finite field discrete logarithm problem) will be broken; if the ring signature is still constructed based on number theory, the security of the ring signature cannot be guaranteed in the quantum era. In recent years, a new cryptosystem based on lattice theory [10,11,12] has become a research topic for the post-quantum cryptography era because of its advantages of better progressive efficiency, parallelism, simple operation, resistance to quantum attacks, and the existence of worst-case random instances. In 2018, Wang et al. [13] presented an anti-quantum ring signature scheme without trapdoors; their scheme adopted the Gaussian “tail-cut” factor, which leads to a relatively long signature length. Torres et al. [14] put forth the first lattice-based one-time linkable ring signature in the random oracle model, which uses the rejection sampling technique to make the distribution of the output signature independent of the distribution of the private key of the signature, thus further improving the efficiency of signature generation. Torres et al. [15] extended the scheme of [14] and proposed a ring signature scheme supporting multiple inputs and multiple outputs, which is more practical. Cui et al. [16] proposed a lattice-based ring signature scheme and vehicular ad hoc network (VANET) privacy preservation; the scheme has high-level security and traceability while ensuring anonymity and is a ring signature scheme based on the hardness problem, which can effectively solve the privacy protection problem in VANETs. Combining the lattice signature and the ring signature, Lui et al. [17] presented a double authentication prevention scheme, which provides secure authentication but lacks full anonymity. In addition, the above schemes demonstrate that the message is transmitted securely from the sender to the receiver and can only be received securely by the receiver. Subsequently, Esgin et al. [18] solved several problems in transferring the design idea of Kohlweiss et al. [19] to lattices; they designed a one-to-many protocol based on the SIS problem [20] in modular lattices and constructed a ring signature scheme with a logarithmic level signature size. Feng et al. [21] proposed a general design framework of a traceable ring signature and constructed a lattice traceable ring signature scheme based on Stern’s protocol. This scheme utilizes techniques of preimage sampling and rejection sampling, and the generation of a key using a trapdoor generation algorithm. It also provides secure authentication, but the efficiency of the scheme is limited by the use of non-interactive zero-knowledge proofs.

### 1.2. Motivation

In huge networks, the privacy of communication is very important. If a legal member of the group securely transmits a message and the message is incorrectly modified by a malicious user, the consequences may affect other users. Owing to the existence of these malicious operations, it is necessary to provide an efficient and secure mechanism to strengthen privacy protection. Although some schemes provide necessary privacy protection, there are many difficulties in distinguishing the malicious operations of authorized users and unauthorized users. A ring signature can hide the signer’s identity from a group, which can better solve these issues.

In this paper, an efficient and secure anti-quantum ring signature scheme is proposed in combination with lattice-based cryptography and ring signature. It helps to verify information and protect the user’s identity privacy. On one hand, most of the proposed lattice-based ring signature schemes are mainly based on two types of problems: small integer solution (SIS) problems and learning with errors (LWEs); they all have an important characteristic in that the time spent solving the two kinds of problems is equivalent to the time spent solving the worst-case hardness problem. On the other hand, in our scheme, our sample is bimodal, having two centers at Se and −Se, namely, $D_{\sigma }^{m}$ (the distribution can be scaled up to $D_{Se,\sigma }^{m}$ or $D_{-Se,\sigma }^{m}$) is under the bimodal distribution. Since our scheme does not adopt the Gaussian “tail-cut” factor, the sampling process can produce shorter signatures. Furthermore, we adopt the encoding function $F:{\left\{0,1\right\}}^{\kappa }\to B_{\eta }^{n}$ to map $h:{\left\{0,1\right\}}^{*}\to \{v:v\in {\left\{-1,0,1\right\}}^{n},{\left\|v\right\|}_{1}\leq \kappa \}$ to $B_{2q}^{n}$ (where $B_{2q}^{n}$ denotes a set of binary vectors of length n and weight η=2q,η that is constant). This method can greatly speed up the signature and verification. 

### 1.3. Our Contribution

As privacy protection is a significant concern, this paper proposes a ring signature scheme based on anti-quantum lattice-based cryptography to solve the vulnerability of the existing schemes to quantum attacks. The ring signature scheme is designed based on the lattice assumption and can support anti-quantum security. The specific research contents include:(1)Combining lattice-based cryptography with a ring signature, we construct a secure lattice-based ring signature under the random oracle model. The proposed scheme satisfies unconditional anonymity and unforgeability. The unforgeability of the proposed ring signature scheme is reduced to the difficult assumption of the small integer solution (SIS) on the lattice.(2)Our scheme also provides a certain degree of unconditional anonymity for ring members and ensures signature unforgeability.(3)We give a detailed performance analysis and provide applications of our scheme in VANETs, and the results show that our scheme is significantly better than the ongoing schemes. Our scheme satisfies security requirements in VANETs.


### 1.4. Outline

The rest of the paper is organized as below. Some preliminaries, such as assumptions and lemmas, are introduced in Section 2. The security model and the architecture of our proposed scheme are described in Section 3 and Section 4, respectively. The correctness and security analysis are provided in Section 5. The performance evaluation is provided in Section 6. We give the related applications in VANETs in Section 7 and present an extension of the scheme in Section 8. Finally, the conclusions are given in Section 9.

## 2. Preliminaries

### 2.1. Notations

If X is a set, then x→X means the entity of picking uniformly random x in X. Let D be a Gaussian distribution and PPT be probabilistic polynomial time; ℝ and ℤ denote real numbers and integers, respectively. ℝ or ℤ is named by lower-case letters (e.g., x) and matrices by bold upper-case letters (e.g., A); AT is the transposition of A. “||“ means the concatenation of strings or matrix columns; vectors are in column form. *negl* (*n*) means a negligible function, and a function ω(f(n)) denotes ω(f(n)) grows faster than cf(n) with any constant c>0. For any matrix X∈ℝn×k, we use s1(X)=max||u||=1||Xu|| to denote the largest singular value (also known as the spectral norm) of X.

### 2.2. Lattices and Lattice Problems

Given m to be linearly independent vectors B=(b1,⋯,bm)∈ℝm×m, an m-dimensional lattice Λ is defined as Λ=L(B)=Bc={∑i=1mbici,ci∈ℤm}, where Λ=L(B) is a basis of B.

For m≥n≥1 and q≥2, a matrix A∈ℤqn×m, the lattice is defined as Λq⊥(A)={e∈ℤm,Ae=0modq}, and Λqu(A)={e∈ℤm,Ae=umodq}. Thus, Λqu(A) is obviously a co-set of Λq⊥(A); namely, Λqu(A)=t+Λq⊥(A). Where t is an arbitrary solution (over ℤqm ) of the equation Ae=umodq, this is the integer lattice called a q-ary lattice.

Let L be a subset of ℤm. For any vector c∈ℝm and any positive parameter δ∈ℝ, let ρδ,c(x)=exp(−πx−c2δ2) be a Gaussian-shaped function on ℝm with center c and parameter δ. Next, for every y∈L, we set ρδ,c(L)=∑x∈Lρδ,c(x) to be the sum of ρδ,c(x) over L with parameters (δ,c) and DL,δ,c(y)=ρδ,c(y)ρδ,c(L). For simplicity, ρδ,0 and DL,δ,0 are abbreviated as ρδ and DL,δ, respectively.

Here, we recall the shortest vector problem (SVP) over lattices. For a lattice basis B and an approximation factor γ, its goal is to find the shortest non-zero vector in a lattice L(B).

### 2.3. Hard Problems for q-ary Lattices

The security of our proposed scheme rests on the following hardness problems that cannot be solved in polynomial time with non-negligible advantage. The related problem is described as follows.

**Definition** **1.**(*SIS problem): The SIS problem is given* (m,q,β) *and* A∈ℤqn×m; *its goal is to compute a non-zero vector* x∈ℤqm *such that* Ax=0modq *with* x≤β.*Ajtai [22] first showed that the SIS problem is hard on average. Later, Micciancio et al. [23] formalized its notion and determined that the SIS problem is regarded as a worst-case hard lattice problem. Micciancio et al., showed that solving the average-case SIS problem was reduced to worst-case, approximating the SVP within certain* β⋅O˜(n) *factors*.

**Lemma** **1.**[24] *For any* k≥1*, if* Pr[s>kσm:s←Dσn]<kne12(1−2k2) *holds, this means* Pr[|s,r|>r:s←Dσn]<2er22ν2σ2*, where* ν∈ℝn *and* σ,r>0.

**Lemma** **2.**[24] *If* Pr[Dσn/Dν,σn<eβ12+12β2:s←Dσn]=1−2−100 *holds for any vector* s∈ℤn *and* σ=βν,β>0.

**Lemma** **3.***For any matrix* A∈ℤqn×m *and* S∈{−d,…,d}m×k*, there is another different* S′∈{−d,…,d}m×k *that satisfies* AS=AS′modq *with a probability not beyond* 1−2−100*, where* m>64+nlogq/log(2d+1).

### 2.4. Chameleon Hash Function

A construction of chameleon hash consists of the following algorithms CHF = (HGen, Hash, Col).

HGen: On input, a security parameter λ outputs (hk,td)←HGen(λ), where the hash key is hk and the trapdoor td.

Hash: On input, the hash key hk and vectors μ and r output the hash value h←Hash(hk,μ,r).

Col: On input, the trapdoor td, $r^{\prime }$ and a message $\mu^{\prime }$, output $r^{\prime }\leftarrow Col^{-1}(td,\mu^{\prime },\mu ,r)$ such that $Hash(hk,\mu^{\prime },r^{\prime })=Hash(hk,\mu ,r)$.

Chameleon hash function supports the property of enhanced collision resistance, which was applied to the design of our scheme.

## 3. System Models

### 3.1. Basic Model

A basic model in our proposed scheme is illustrated in Figure 1, where $P_{1},P_{2},\ldots ,P_{k}$ denotes ring members. In this model, new ring members and other members will form a common ring; we call a group of possible signers a ring. The ring members can create the actual signature, and other ring members who cannot generate an efficient signature are called non-signers. For example, in a network model of VANETs, most information, such as beacon messages periodically broadcast by vehicles and public messages broadcast by roadside units, do not need to be kept secret, but these messages are associated with responsibility. Before using the message, it is necessary to verify whether the message comes from a legitimate network member and to check the authenticity of the message, so signature technology is required. The vehicle will use the ring member to sign and issue follow-up messages so as to effectively hide its real identity on the premise of ensuring the authenticity of the messages and to realize anonymous communication in the VANETs. Applying ring signature technology helps vehicles construct a ring with nearby vehicles through roadside facilities, and the real identity of the signer can be identified according to the signed message so as to realize the unconditional anonymous communication of vehicles in VANETs.

As the main method of resisting quantum attack, lattice-based cryptography has been widely considered. In addition, ring signature has good anonymity and unforgeability, so we believe that the ring signature scheme based on the lattice hard problem can effectively solve the privacy protection problem in practical applications (such as VANETs).

### 3.2. Threat Model

For our proposed lattice-based ring signature scheme, we considered a widely accepted threat model [2]. In terms of the model, an adversary A cannot distinguish that the member of a ring created a given signature among the communicated entities in the application environment. Furthermore, any communicating entities (unauthorized users or attackers) cannot output signatures. Specific details are described as follows.

Anonymity. The following game between challenger C and adversary A is used to define the anonymity of the ring signature scheme:
(1)A creates a group of public parameters P=(L,n,m,q), a ring $R=(pk_{1},\ldots ,pk_{n})$, two secret keys ($sk_{i_{0}},sk_{i_{1}})$, and a message μ.(2)A is permitted to make ring-signing queries and corruption queries. C responds with $\sigma_{L}(\mu )=Sign(pk_{s},sk_{s},R,\mu )$ as a ring-signing query. The signer of an index s performs a corruption query. Finally, C sends $sk_{s}$ to A.(3)A requests a challenge to C with the values $\left(i_{0},i_{1},R,\mu \right)$; C calculates two challenge signatures.$\sigma_{i_{0}}=Ring-sign(P,sk_{i_{0}},R,\mu )$ and $\sigma_{i_{1}}=Ring-sign(P,sk_{i_{1}},R,\mu )$; C responds to A with $\sigma_{i_{0}}$, $\sigma_{i_{1}}$.(4)A responds a guess $b^{\prime }$ and wins the game if $b^{\prime }=b$.


Unforgeability. To enable signature verification:

$Sign-Verify(R,\mu ,\sigma_{L}(\mu ))=1$ forgery is implemented when an unauthorized user obtains the private key from $R=(pk_{1},\ldots ,pk_{n})$ or a ring member that has previously signed a message. The unforgeability with insider corruption is defined as the following game between a challenger C and an adversary A:
(1)A creates a group of public parameters P=(L,n,m,q), a ring $R=(pk_{1},\ldots ,pk_{n})$, two secret keys ($sk_{i_{0}},sk_{i_{1}})$, and a message μ.(2)A is permitted to make ring-signing queries and corruption queries. C responds with $\sigma_{L}(\mu )=Sign(pk_{s},sk_{s},R,\mu )$ as a ring-signing query. The signer of an index s performs a corruption query. Finally, C sends $sk_{s}$ to A.(3)A sends the result $\left(R,\mu^{*},\sigma_{L}{\left(\mu \right)}^{*}\right)$ to the challenger, and A is considered as successful if $Sign-Verify(R,\mu^{*},\sigma_{L}{\left(\mu \right)}^{*})=1$, where $\mu^{*}\notin \mu$.


## 4. The Proposed Scheme Description

In this section, to facilitate the description of our scheme, we use a bimodal Gaussian distribution as a major building block for our ring signature scheme. The aim is to make sampling rejection more effective, and the procedures for rejecting sampling are illustrated in [24].

With the technique employed in [25], we present a ring scheme over lattices and prove its security under the SIS problem. The relevant steps are as follows:

Key generation: Given a security parameter, and some other parameters n,m,q,i,j, let Ai∈ℤ2qn×m and Si∈ℤ2qm×n be public/private keys of the user with index i, respectively, such that key pairs meet AiSi=qInmod2q (where i∈L={1,2,…,n}, Si is invertible). Let a hash function be h:{0,1}*→{v:v∈{−1,0,1}n,v1≤κ} and nearly injective mapping be F:{0,1}κ→B2qn (B2qn denotes a set of binary vectors of length n and weight 2q). For A^=(A1,A2,…,An),Ai∈ℤ2qn×m and S^=(S1,S2,…,Sn),Si∈ℤ2qn×m, this is (pk,sk)=(A^,S^). The system publishes P=(L,pk,h,F,n,m,q). The relevant details are shown in the following Algorithm 1.

**Algorithm 1** KeyGen algorithm**Input:** A security parameter λ**Output:** The public parameters P1: Let L={1,2,…,n} and q be a prime number;2: Define three sets Dσ1m,Dσ2m,Dσ3m, hash function h:{0,1}*→{v:v∈{−1,0,1}n,v1≤κ}, and nearly injective mapping F:{0,1}κ→B2qn;3: Set A^=(A1,A2,…,An),Ai∈ℤ2qn×m and S^=(S1,S2,…,Sn),Si∈ℤ2qm×n such that AiSi=qInmod2q;4: Set (pk,sk)=(A^,S^);5: Output P=(L,pk,h,F,n,m,q).

Ring-Sign: On input, a message μ, a long-term key $S_{j}$, a ring of n members with public keys $\hat{A}=(A_{1},A_{2},\ldots ,A_{n})$, a user i selects uniform value $k_{i}\leftarrow D_{\sigma_{1}}^{m}$ and calculates $x_{i}=A_{i}y_{i}\mathrm{mod}2q$ with the random vector $y_{i}\leftarrow D_{\sigma_{2}}^{m}$, and outputs the signature $\sigma_{L}(\mu )$ as illustrated in Algorithm 2 of the message μ. Then, the user i performs the following computations:
(1)For all i∈L, calculate $h_{i}=x_{i}+A_{i}k_{i}\mathrm{mod}2q$, where L={1,2,…,n}.(2)Calculate $e=(\sum_{i\in L}h_{i}\mathrm{mod}2q,\mu )$ and $\tilde{e}=F(e)$.(3)Pick a random bit b∈{0,1}; calculate $s_{j}=y_{j}+k_{j}+{\left(-1\right)}^{b}S_{j}\tilde{e}$, where i=j.(4)For i≠j, compute $s_{i}=y_{i}+k_{i}\mathrm{mod}2q$.(5)Publish $\sigma_{L}(\mu )=({\left\{s_{i}\right\}}_{i\in L=\{1,2,\ldots ,n\}},e)$.


**Algorithm 2** Ring-signing algorithm**Input:** A message μ, a long-term key Sj, public keys A^=(A1,A2,…,An)**Output:** The signature σL(μ)1: Calculate hi=xi+Aikimod2q, xi=Aiyimod2q, where i∈L={1,2,…,n};2: Calculate e=(∑i∈Lhimod2q,μ) and e˜=F(e);3: For i∈L={1,2,…,n} and i≠j, compute si=yi+kimod2q;4: Pick b∈{0,1}; compute sj=yj+kj+(−1)bSje˜, where i=j;5: Continue the next steps with probability 1Mexp(−Sj⋅e22σ2)cosh(sj,Sj⋅eσ2),otherwise **Restart**;6: Output σL(μ)=({si}i∈L={1,2,…,n},e).

Ring-Verify: Given a signature $\sigma_{L}(\mu )$, a message μ, and a bit b, the algorithm outputs a response and answers: accept or reject (as illustrated in Algorithm 3). The signature $\sigma_{L}(\mu )$ can be checked and only accepted under the following conditions: ${\left\|s_{i}\right\|}_{2}\leq B_{2}$ and ${\left\|s_{i}\right\|}_{\infty }\leq q/4$ for 1≤i≤n, where $B_{2}$ is the valid bounds [26].
(1)$s_{i}\leftarrow D_{\sigma_{3}}^{m}$(2)$e=(\sum_{i\in L}A_{i}s_{i}+q\tilde{e}\mathrm{mod}2q,\mu )$


If the above verifications hold, the signature is valid and the verifier outputs 1; otherwise, it outputs 0.

**Algorithm 3** Ring-verify algorithm**Input:** The signature σL(μ); public keys A^=(A1,A2,…,An)**Output:** Accept or Reject1: **If**
si←Dσ3m, then **continue**;2:   else if si2≤B2, then **continue**;3:   else if si∞≤q/4, then **continue**;4: **else if**
e=(∑i∈LAisi+qe˜mod2q,μ), then **Accept**,  **else Reject**;5: Output Accept or Reject.

**Theorem** **1.***Define* $B_{2}=\eta \sigma \sqrt{m}$ *and* $q/4>\sqrt{\left(\lambda +1)In2+2In(m\right)}\sigma$ *and a signature* $\sigma_{L}(\mu )$. *These parameters are created based on Algorithm 2. Then, the output of Algorithm. 3 outputs accept with probability* 1−λ/2 *if* $\sigma_{L}(\mu )$ *is valid*.

**Proof.** In terms of Lemma 2 and Lemma 3, we find that the bound on Euclidean norm is $B_{2}=\eta \sigma \sqrt{m}$, and, for any η>1, there is a probability $\mathrm{Pr}[{\left\|s_{i}\right\|}_{2}\geq \eta \sigma \sqrt{m}]>1-\lambda /2$. According to Lemma 2 and Lemma 3, we find that the bound on infinity norm is ${\left\|s_{i}\right\|}_{\infty }\leq q/4$. In fact, it satisfies the following conditions $q/4>\eta \sigma >\sqrt{\left(\lambda +1)In2+2In(m\right)}\sigma$ unless its probability is λ/2. □

## 5. Correctness and Security Analysis

### 5.1. Correctness

The correctness of the signature can be well verified. In fact, the signer outputs the form of the signature $\sigma_{L}(\mu )=({\left\{s_{i}\right\}}_{i\in L=\{1,2,\ldots ,n\}},e)$, where $s_{i}\leftarrow D_{\sigma_{3}}^{m}$. The signature is valid if the following details are true:
(1)$$\begin{array}{l} \sum_{i\in L}A_{i}s_{i}+q\tilde{e}=\sum_{i\in L,i\neq j}(A_{i}y_{i}+A_{i}k_{i})+q\tilde{e}+A_{j}s_{j}\\ \;=\sum_{i\in L}x_{i}+\sum_{i\in L,i\neq j}A_{i}k_{i}+A_{j}({\left(-1\right)}^{b}S_{j}\tilde{e}+k_{j})+q\tilde{e}\\ \;=\sum_{i\in L}x_{i}+\sum_{i\in L}A_{i}k_{i}+{\left(-1\right)}^{b}q\tilde{e}+q\tilde{e}\\ \;=\sum_{i\in L}h_{i}\mathrm{mod}2q \end{array}$$


Therefore, $e=h(\sum_{i\in L}h_{i}\mathrm{mod}2q,\mu )$.

### 5.2. Security Analysis

**Lemma** **4.***For the tuple* $\left(i_{0},i_{1},R,\mu \right)$*, a message* μ*, the ring* $R=(A_{1},A_{2},\ldots ,A_{n})$*, and* $i_{0}$ *and* $i_{1}$ *are indices with* $A_{i_{0}},A_{i_{1}}$*. If the SIS problem is hard,* $\sigma_{i_{0}}\leftarrow Sign(sk_{i_{0}},R,\mu )$ *and* $\sigma_{i_{1}}\leftarrow Sign(sk_{i_{1}},R,\mu )$ are computationally indistinguishable.

**Proof.** Let $Y_{b,P,sk_{i_{b}},\mu }$ be some uniform distribution in ring R; there is a random variable describing the output of $Ring-sign(b,sk_{i_{b}},R,\mu )$ with ring R, where $sk_{i_{b}},\mu$ denotes a group of arbitrary inputs and b∈{0,1}. If the domains of the above variables are different, it means that the signature fails. Then, we have
(2)$$\Delta (Y_{0,P,sk_{i_{0}},\mu }-Y_{1,P,sk_{i_{1}},\mu })=n^{-\omega (1)}$$
Therefore, $\sigma_{i_{0}}$ and $\sigma_{i_{1}}$ have the same domain distribution within a negligible statistical distance of $\Delta (Y_{0,P,sk_{i_{0}},\mu }-Y_{1,P,sk_{i_{1}},\mu })$, and this means that $\sigma_{i_{0}}$ and $\sigma_{i_{1}}$ are computationally indistinguishable. □

**Theorem** **2.**(Anonymity): *Our ring signature scheme is anonymous under the hardness of SIS*.

**Proof.** To prove the security of our scheme, there are the following two cases: ① Signatures created by ring signers and non-signers are entirely indistinguishable. ② The attacker cannot obtain the private key of the signer by utilizing the public key of all ring members in polynomial time. □

On one hand, in Algorithm 2, the signer using its private key generates the tuples $\sigma_{L}(\mu )=({\left\{s_{i}\right\}}_{i\in L=\{1,2,\ldots ,n\}},e)$. For 1≤i=j≤L, $s_{j}=y_{j}+k_{j}+{\left(-1\right)}^{b}S_{j}\tilde{e}$; whereas the other part is produced utilizing public keys of the ring non-signer, i.e., $s_{i}=y_{i}+k_{i}$, where 1≤i≠j≤L. In the meantime, we rewrite this part $s_{i}=(y_{i}+{\left(-1\right)}^{b}S_{i}\tilde{e})+(k_{i}-{\left(-1\right)}^{b}S_{i}\tilde{e})\mathrm{mod}2q={y^{\prime }}_{i}+{k^{\prime }}_{i}\mathrm{mod}2q$, where

${y^{\prime }}_{i}=y_{i}+{\left(-1\right)}^{b}S_{i}\tilde{e}$,${k^{\prime }}_{i}=k_{i}-{\left(-1\right)}^{b}S_{i}\tilde{e}$, which means that the probability of distinguishing between the uniformly created sample and the $s_{i}={y^{\prime }}_{i}+{k^{\prime }}_{i}\mathrm{mod}2q$ sample is negligible. Thus, in the attacker’s view, the signatures created by the ring signer and the ring non-signer are indistinguishable.

On the other hand, assume that there exists an adversary A generating a forgery $\sigma_{L}{\left(\mu \right)}^{*}$ with probability $\varepsilon^{\prime }$. We build an algorithm C that utilizes A to solve the instance of the SIS problem with probability ε. To respond to A’s queries, C maintains three lists h, F, and G, which are initialized to null and store tuples of values. Then, C interacts with A as follows:

In the Setup phase, C produces Ai∈ℤ2qn×m and Si∈ℤ2qm×n. C stores the tuple (i,Ai,Si), where i∈L={1,2,…,n} in list G and the related parameters (A1,A2,…,An) are given to A. In the query phase, C responds to the three queries of G as below:

Hash queries: C submits a random value $y_{i}\leftarrow D_{\sigma_{2}}^{m}$ to A and stores $\left(y_{i},h_{i}\right)$ in h-list. In addition, C picks a random value e to A and stores it in F-list.

Corruption queries: C searches for the tuple $\left(i,A_{i},S_{i}\right)$ in G-list and responds to A with $S_{i}$.

Signing queries: C calculates the signature with the below steps:
(1)For all i∈L={1,2,…,n}, calculate $h_{i}=x_{i}+A_{i}k_{i}\mathrm{mod}2q$, $x_{i}=A_{i}y_{i}\mathrm{mod}2q$, where i∈L={1,2,…,n}.(2)Calculate $e=(\sum_{i\in L}h_{i}\mathrm{mod}2q,\mu )$ and $\tilde{e}=F(e)$.(3)Pick a random bit b∈{0,1}; calculate $s_{j}=y_{j}+k_{j}+{\left(-1\right)}^{b}S_{j}\tilde{e}$, where i=j.(4)For i≠j, compute $s_{i}=y_{i}+k_{i}\mathrm{mod}2q$.(5)Publish $\sigma_{L}(\mu )=({\left\{s_{i}\right\}}_{i\in L=\{1,2,\ldots ,n\}},e)$.


C returns the signature $\sigma_{L}(\mu )$ to A.

*Analysis.* In a way, A performs the Ring-signing with $\left(i_{0},i_{1},R,\mu \right)$ and public key $pk_{i_{0}},pk_{i_{1}}$ over ring R; C retrieves the tuple $\left(y_{i},h_{i}\right)$ in h-list. C calculates the challenge signature $\sigma_{L}{\left(\mu \right)}^{*}$ and sends $\sigma_{L}{\left(\mu \right)}^{*}$ to A. Finally, A outputs a guess b∈{0,1}. From the viewpoint of A, the behavior of C is statistically close to the one provided by the real adaptive security experiment. We find that the ring members calculate $e^{*}=(\sum_{i\in L}h_{i}\mathrm{mod}2q,\mu^{*})$, ${\tilde{e}}^{*}=F(e^{*})$, $s_{j}{}^{*}=y_{j}+k_{j}+{\left(-1\right)}^{b}S_{j}{\tilde{e}}^{*}$(for i=j), and $s_{i}{}^{*}=y_{i}+k_{i}$(for i≠j). C outputs $\sigma_{L}{\left(\mu \right)}^{*}=({\left\{s_{i}{}^{*}\right\}}_{i\in L=\{1,2,\ldots ,n\}},e^{*})$ as a signature of $\mu^{*}$.

If A provides another success probability in distinguishing between $i_{0}$ and $i_{1}$ with a non-negligible probability, it seems to contradict Lemma 4. Thus, we declare that the advantage of A guessing the correct information in the simulated anonymous game is negligible.

**Theorem** **3.****(Unforgeability)**: *Our ring signature scheme is unforgeable by insider corruption assuming that the SIS problem is hard*.

**Proof.** To prove the security of our scheme, the following two cases were considered: ① The attacker cannot break the security assumption of the scheme. ② The attacker cannot find the collision in the anti-collision hash function. Regarding the above two problems, we start the proof of this part. □

Assume that there exists an adversary A that creates a forgery $\sigma_{L}{\left(\mu \right)}^{*}$ with probability $\varepsilon^{\prime }$. We build an algorithm C, which utilizes A to solve the instance of the SIS problem with probability ε. Then, C interacts with A as follows: 

C picks i∈L={1,2,…,n} and guesses the size of the challenge ring. In addition, C selects a vector t=(t1,t2,…,tn). To respond to A’s hash queries and signing queries in the random oracle, C will maintain three lists, h, F, and G, which are initialized to be empty and will store tuples of values. For any i∈L={1,2,…,n} and i∉t, C produces Ai∈ℤ2qn×m and Si∈ℤ2qm×n. C stores the tuple (i,Ai,Si), where i∈L={1,2,…,n} in list G and the relevant parameters (A1,A2,…,An) are sent to A.

Query Phase: C responds to adaptive queries from A on any message μ as follows:

Hash queries: C submits a random value $y_{i}\leftarrow D_{\sigma_{2}}^{m}$ to A and stores $\left(y_{i},h_{i}\right)$ in h-list. In addition, C sends a random value e to A and stores it in F-list.

Corruption queries: C searches for the tuple (i,Ai,Si) in G-list and responds to A with Si.

Signing queries: C calculates the signature $\sigma_{L}(\mu )=({\left\{s_{i}\right\}}_{i\in L=\{1,2,\ldots ,n\}},e)$ for the requested message and returns the signature $\sigma_{L}(\mu )$ to A.

Namely, A receives signature $\sigma_{L}(\mu )$ and computes as follows:
$$\begin{array}{l} \mathrm{Pr}[Forge-sign_{A}(\lambda )=1]=\mathrm{Pr}[Forge-sign_{A}(\lambda )=1\cap Hash-collision_{A}(\lambda )]\\ +\mathrm{Pr}[Forge-sign_{A}(\lambda )=1\cap {\bar{Hash-collision}}_{A}(\lambda )]\\ \leq \mathrm{Pr}[Hash-collision_{A}(\lambda )]+\mathrm{Pr}[Forge-sign_{A}(\lambda )=1\cap {\bar{Hash-collision}}_{A}(\lambda )] \end{array}$$
where $\mathrm{Pr}[Forge-sign_{A}(\lambda )=1]$ represents that A can find the probability of collision in the hash function, and $\mathrm{Pr}[Hash-collision_{A}(\lambda )]$ denotes the probability of creating a forgery of the signature.

Challenge: Finally, A outputs a forgery signature $\sigma_{L}{\left(\mu \right)}^{*}=({\left\{s_{i}{}^{*}\right\}}_{i\in L=\{1,2,\ldots ,n\}},e^{*})$. If $R^{*}=R$, C aborts. Otherwise, C skips the tuple $\left(y_{i},h_{i}\right)$ in h-list and outputs $\sigma_{L}{\left(\mu \right)}^{*}$ as a collision of μ.

*Analysis.* To some extent, the view of A in the adaptively chosen message attack is the same as the view provided by C. For each distinct query $h_{i}$ and F, C returns $e=(\sum_{i\in L}h_{i}\mathrm{mod}2q,\mu )$ and $\tilde{e}=F(e)$. Through the unified output characteristics of the constructed hash function, it is the same as a uniform random value of $\left(\sum_{i\in L}h_{i}\mathrm{mod}2q,\mu \right)$ in the real environment. Thus, A outputs a valid forgery $\sigma_{L}{\left(\mu \right)}^{*}$ negligibly close to ε.

Suppose A creates a response $\left(s_{i},\tilde{e}\right)$ in the hash query, which is $h(A_{i}s_{i}+q\tilde{e},\mu )=h(A_{i}s_{i}{}^{*}+q{\tilde{e}}^{*},\mu^{*})$ f or two different signatures $\left(s_{i},e,\mu \right)$ and $\left(s_{i}{}^{*},e^{*},\mu^{*}\right)$. From the above signature, there is a hash collision if $\mu^{*}\neq \mu$ or $A_{i}s_{i}+q\tilde{e}\neq A_{i}s_{i}{}^{*}+q{\tilde{e}}^{*}$ holds. However, this is impossible according to the characteristics of hash function. Thus, $\mu^{*}=\mu$ or $A_{i}s_{i}+q\tilde{e}=A_{i}s_{i}{}^{*}+q{\tilde{e}}^{*}$. The following equation holds:
$$A_{i}(s_{i}-s_{i}{}^{*})=0\mathrm{mod}2q$$
since ${\left\|s_{i}\right\|}_{\infty }\leq q/4$ and ${\left\|s_{i}{}^{*}\right\|}_{\infty }\leq q/4$, this is $s_{i}-s_{i}{}^{*}\neq 0\mathrm{mod}2q$, where the condition on $\left\|s_{i}-s_{i}{}^{*}\right\|$ is $2B_{2}$. This means that the SIS problem can be solved.

According to the proof of [26], suppose C publishes a forgery $e_{t}$ to the forger as a response. Then, we set a ring signature $\left(s_{t},e_{t}\right)$ for a message μ. Therefore, for any different values $({e^{\prime }}_{1},{e^{\prime }}_{2},\ldots ,{e^{\prime }}_{\rho })\leftarrow T^{k}$ and $b\leftarrow T^{n}$. The algorithm of the same time-complexity as the forger observes $(e_{t}-{e^{\prime }}_{t})\neq 0\mathrm{mod}2q$ with probability is:
(3)$$\mathrm{Pr}[(e_{t}-{e^{\prime }}_{t})\neq 0]=(\varepsilon -\frac{1}{T_{k}^{n}})(\frac{\varepsilon -\frac{1}{T_{k}^{n}}}{b}-\frac{1}{T_{k}^{n}})$$


Next, C produces a response $s_{j}$ to A. We assume that there is a ring signature $\left(e^{*},s_{j}{}^{*}\right)$ of $\mu^{*}$, and A picks the various $\left(s_{1}{}^{*},s_{2}{}^{*},\ldots ,s_{n}{}^{*}\right)$. Since $\tilde{e}=F(e)$, ${\tilde{e}}^{*}=F(e^{*})$$A_{j}s_{j}+q\tilde{e}\neq A_{j}s_{j}{}^{*}+q{\tilde{e}}^{*}$; this is $A_{j}(s_{j}-s_{j}{}^{*})\neq q({\tilde{e}}^{*}-\tilde{e})$. Since ${\tilde{e}}^{*}-\tilde{e}\neq 0\mathrm{mod}2$, this is $s_{j}-s_{j}{}^{*}\neq 0\mathrm{mod}2q$. Furthermore, we find ${\left\|{\tilde{e}}^{*}-\tilde{e}\right\|}_{\infty }\leq q/2$; this implies $v=s_{j}-s_{j}{}^{*}\mathrm{mod}2q$. Thus, $A_{i}v=0\mathrm{mod}2q$; $\left\|v\right\|\leq 2B_{2}$. It means that we can obtain the solution to the SIS problem.

In other words, so long as A successfully breaks through the strong unforgeability of our scheme, C can effectively solve the SIS problem. Thus, the probability of successfully solving the SIS problem is negligible.

On the other hand, if a hash collision does not exist in our scheme, that means A generates the forged valid signature on message μ if A finds the private key of C using R. In fact, the hardness of the SIS problem, the problem of $\mathrm{Pr}[Forge-sign_{A}(\lambda )=1\cap {\bar{Hash-collision}}_{A}(\lambda )]$, is the probability of finding a private key by utilizing the corresponding public key, but the case is negligible.

## 6. Performance Evaluation

### 6.1. Parameter Selection

There are some parameters in our ring signature scheme, as illustrated in Table 1, that were chosen in the same way as [14]. They are secure against direct lattice attacks in terms of the algorithm Hermite factor δ, using the value of δ=1.007. In addition, the complexity of the SIS problem should be achieved by appropriate selection of parameters n,m,q,κ, where κ represents the challenge size that meets $2^{\kappa }\cdot \left(\begin{array}{c} n\\ \kappa \end{array}\right)\geq 2^{-100}$. Then, the correctness error of the rejection sampling will be within at most $2^{-100}$. As illustrated in Lemma 1 and Lemma 2, the equation below holds that
(4)$$\frac{D_{\sigma }^{m}(s_{j})}{MD_{\nu ,\sigma }^{m}(s_{j})}=\frac{1}{M\exp(-\frac{{\left\|S_{j}\cdot e\right\|}^{2}}{2\sigma^{2}})\cosh(\frac{\langle s_{j},S_{j}\cdot e\rangle }{\sigma^{2}})}\leq 1$$


Thus, we set $M=\frac{1}{\exp(-\frac{{\left\|S_{j}\cdot e\right\|}^{2}}{2\sigma^{2}})\cosh(\frac{\langle s_{j},S_{j}\cdot e\rangle }{\sigma^{2}})}$. For $\sigma =12\left\|S_{j}\cdot e\right\|$ from Lemma 2, M = 1.0027.

Next, we analyzed the parameters in the proposed scheme that satisfied the following conditions, as shown in Table 1.

### 6.2. Efficiency Analysis

We analyzed the performance of elements of our scheme, such as the public key size, private key size, and signature size; the related details of the efficiency analysis are shown in Table 2. Then, we computed the signature size for the different security levels, such as 100, 128, 256, and 512 bits, and the results are shown in Table 2.

As shown in Figure 2, the signature size of our proposed scheme increased rapidly as the security level increased. However, the increase in signature size in this scheme was small. The size of the signature remained stable regardless of the security level. In addition, the proposed scheme resisted quantum attacks. Thus, with the advent of quantum computers, lattice-based cryptography will gradually be integrated into practical scenarios in the future quantum age.

### 6.3. Performance Comparison

In our scheme, the ring contains n members; the total space complexity is O(n). Suppose that the time of multiplication operation is $T_{mult}$, non-interactive zero-knowledge proof operation is $T_{n}$, and hash operation is $T_{h}$. We provided a comparison of our scheme and the relevant schemes in terms of ring-sign, ring-verify costs, and signature length, as illustrated in Table 3. The addition was neglected in our scheme.

From Table 3, we found that our scheme was highly efficient, and the computational costs of ring signature generation and verification were lower than those in the literature [16,17,21,27,28]. Next, we performed the functionality comparison of the related schemes, as shown in Table 4. In Table 4, we compared the performance of our proposed scheme with the current prevailing schemes, i.e., Wang et al. [11], Cui et al. [16], Liu et al. [17], Mundhe et al. [27], Feng et al. [21], Han et al. [28], Cai et al. [29]. The scheme of Cai et al. [29] could not resist quantum attack. Mundhe et al. [27], Han et al. [28], and our scheme satisfied both unconditional anonymity and strong unforgeability.

We performed the same scenario as the literature [30] and applied the relevant operation parameters. Next, we evaluated the performance of the related schemes under the same quantum environment. In Figure 3, we provided the ring-sign and ring-verify operation times of the relevant schemes for the numbers of ring members. In addition, our scheme was a ring signature scheme without a trapdoor; we can confirm that our scheme functioned better than the other schemes.

## 7. Sharper Ring Rignatures

We presented another extension of the scheme that achieves faster key generation, signature, and verification than most (traditional or lattice-based) signature schemes. We chose the relevant parameters, including a high-security environment against quantum attacks. The details are described as follows.

Key generation: Given a security parameter λ, and some other parameters n,m,q,i,j, for any integer q, we write ℤq=ℤ/qℤ for simplicity. The ring Rq=ℤq[x]/(xn+1) is isomorphic to ℤq. Let a hash function h:{0,1}*→{v:v∈{−1,0,1}n,v1≤κ}, nearly injective mapping F:{0,1}κ→B2q and i∈L={1,2,…,m}, A=(a1,a2,…,am−1),ai∈Rq, where A∈Rq1×(m−1). Let Aq,i∈Rq1×(m−1) and Sq,i∈Rq(m−1) be public/private keys of the user with index i, respectively, such that key pairs meet Aq,iSq,i=ai (Sq,i is invertible). Let A2q,i=[2A2q,i,q−2ai]∈R2q1×m; this is pk=({A2q,i}i∈L,{Aq,i}i∈L−1,), sk={Sq,i}i∈L. The system publishes P=(L,pk,h,F,n,m,q).

Ring-Sign: On input a message μ, a long-term key $S_{q,j}$, a ring of m members with public keys pk, a user i selects a uniform value $k_{i}\leftarrow D_{\sigma_{1}}^{m}$ and performs the algorithm $x_{i}=A_{2q,i}y_{i}\mathrm{mod}2q$ with the random vector $y_{i}\leftarrow D_{\sigma_{2}}^{m}$, and outputs the signature $\sigma_{L}(\mu )$ of the message μ. Then, the user i performs the following computations:
(1)Set $S_{q,j}^{T}\in R_{q}^{(m-1)\times 1}$ and $S_{2q,j}^{T}=(S_{q,j}^{T},1)\in R_{2q}^{m\times 1}$ such that $A_{2q,i}S_{2q,i}=q$.(2)For all i∈L, calculate $h_{i}=x_{i}+A_{2q,i}y_{i}\mathrm{mod}2q$, where L={1,2,…,m}.(3)Calculate $e=({\left\lfloor \sum_{i\in L}h_{i}\right\rfloor }_{d},\mu )$, where ${\left\lfloor \sum_{i\in L}h_{i}\right\rfloor }_{d}$ denotes high-order bits of $\sum_{i\in L}h_{i}$.(4)Calculate $\tilde{e}\leftarrow F(e)$.(5)Pick a random bit b∈{0,1}; calculate $s_{j}=y_{j}+k_{j}+{\left(-1\right)}^{b}S_{2q,j}\tilde{e}$, where i=j.(6)For i≠j, compute $s_{i}=y_{i}+k_{i}\mathrm{mod}2q$.(7)Publish $\sigma_{L}(\mu )=({\left\{s_{i}\right\}}_{i\in L=\{1,2,\ldots ,n\}},e)$.


Ring-Verify: Given a signature $\sigma_{L}(\mu )$, a message μ, and a bit b, the algorithm outputs a response and answers: accept or reject. The signature $\sigma_{L}(\mu )$ can be checked and is only accepted under the following conditions: ${\left\|s_{i}\right\|}_{2}\leq B_{2}$ and ${\left\|s_{i}\right\|}_{\infty }\leq q/4$ for 1≤i≤n.
(1)$s_{i}\leftarrow D_{\sigma_{3}}^{m}$(2)$e=({\left\lfloor \sum_{i\in L}A_{2q,i}s_{i}+q\tilde{e}\right\rfloor }_{d},\mu )$


If the above verifications hold, the signature is valid, and the verifier outputs 1; otherwise, it outputs 0.

## 8. Applications in VANETs

Vehicular ad hoc networks (VANETs) are a kind of mobile ad hoc network that can intelligently control the entire traffic process and improve traffic efficiency and security. They include two communication modes: vehicle to vehicle (V2V) and vehicle-to-infrastructure (V2I). In VANETs, there exist three types of entities that include the trusted authority (TA), on-board units (OBUs), and roadside units (RSUs), as shown in Figure 4.

TA: TA is responsible for the enrollment of the OBUs and RSUs and produces the system’s public parameters and private key.

OBU: The OBU can share the corresponding traffic information with other vehicles or RSU under the support of the DSRC protocol. Each vehicle is equipped with an OBU. The OBU can send basic information to the RSU and OBUs of other vehicles and verify the received information. Each OBU contains a tamper-proof device (TPD) and a global positioning system (GPS), which ensure that the information stored in it will not be disclosed. The GPS is used to provide geographic location and time information services while driving.

RSU: The RSU is a fixed infrastructure installed along the roadside. The RSU enters VANETs through wireless connection and is managed by the traffic management department through trusted authorization. The RSU verifies the signature immediately after receiving the information from the vehicles. If the signature is valid, the RSU can broadcast the vehicle’s identity information. Otherwise, the RSU discards the relevant information. In addition, the RSU communicates with neighbor RSUs at the same time.

To achieve security authentication, we considered a new privacy protection scheme for VANETs, where the connected vehicles form a common ring with nearby vehicles. In our network model, most information comprised vehicles periodically broadcasting sign messages and RSUs broadcasting public information, but the message was associated with responsibility. Before confirming whether the message came from a legal member of the network, we had to verify it effectively, so we used ring signature technology. The vehicles used the ring signature to sign the subsequent messages so as to effectively hide their real identity under the premise of ensuring the authenticity of the message and to realize anonymous communication in the VANETs. We applied the ring signature to the RSU to help the vehicles quickly form a ring with nearby vehicles.

### 8.1. Experimental Simulation

Without a loss of generality, assume that, in a heavily vehicular area of the city, there are enough vehicles and enough time to form a ring. The time of signature generation is acceptable relative to the time of passing by the base station, which means that our proposed scheme can meet the requirements of composing rings and generating ring signatures. We used the network simulator NS3 [31] to simulate our scheme and employed an Intel Core2 (TM) i5-7300 with 3.4-GHz frequency rate and Windows 10 platform to implement the experiment. We simulated the operation of the vehicle network communication scheme in a real traffic environment. Since the speed is affected by the number of vehicles, we simulated a 1 km-long intersection situation and considered the average speed of the vehicle to be 20 km/h. The RSU was located in the middle of the intersection, the fixed speed was 50 B/s in the network bandwidth, and the transmission bound of the vehicle was 100 m, as illustrated in Table 5. In addition, the area of the simulation was $1\times 1\;{\mathrm{km}}^{2}$, which was controlled by an RSU.

### 8.2. Simulation Results

We evaluated the effectiveness of our proposed scheme from two aspects: end-to-end delay (E2ED) and throughput (THP). E2ED represents the average delay time spent by data packets. THP represents the average number of bits of information transmitted per unit time, as shown in Figure 5 and Figure 6.

Figure 5 and Figure 6 show the simulation results of the experiment. The message authentication delay values were related to vehicle density, where vehicle density represented the number of vehicles within the range of the RSU at a given time. With the continuous increase in vehicles, the scale of the formed ring continued to change. Therefore, with the increase in ring members, the message authentication delay and throughput continued to increase as the vehicle density increased.

## 9. Conclusions

The post-quantum secure ring signature is an important part of post-quantum cryptography and provides a cryptographic tool for user privacy protection in the post-quantum era. Most existing lattice-based ring signature schemes rely on the lattice-based trapdoor function, but the parameters are too large and the efficiency is low, resulting in their inefficiency. In this paper, we proposed a way of building a lattice-based ring signature scheme without a trapdoor. This scheme is a practical lattice-based dynamic ring signature scheme that is suitable for large-scale and scalable application scenarios. Then, we proved its security under the hardness of the SIS problem: the construction satisfied the properties of anonymity and unforgeability. Finally, we applied our scheme to the VANETs, and the simulation results showed that our scheme was feasible. In addition, the development of quantum computers has made an impact on classical cryptography, and the reconstruction of public key cryptography based on the hard problems of anti-quantum computing is the main development direction in the future.

## Figures and Tables

**Figure 1 entropy-23-01364-f001:**
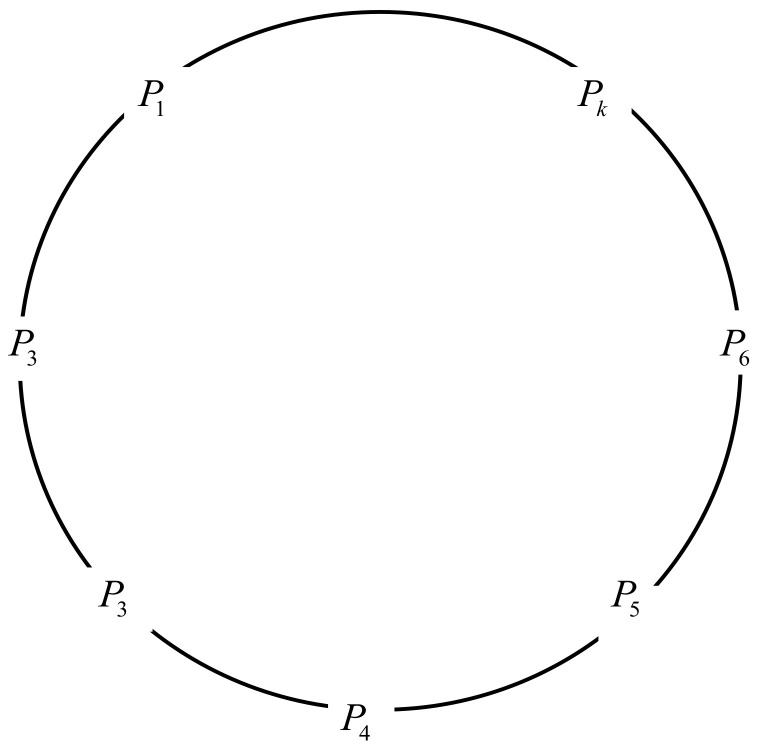
Framework of ring signature.

**Figure 2 entropy-23-01364-f002:**
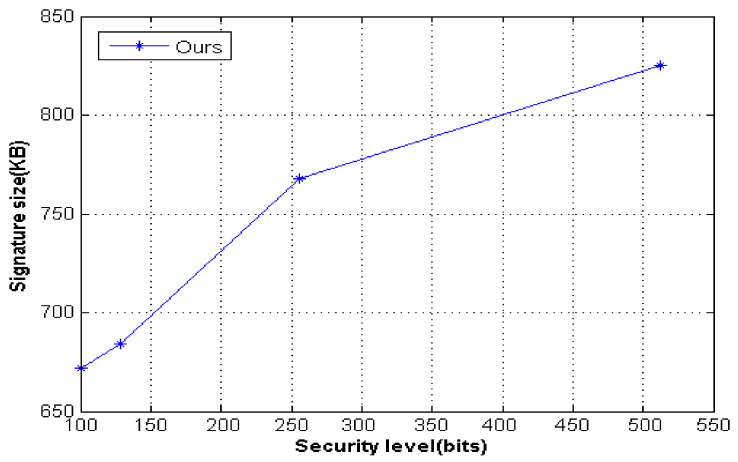
Different security levels of our scheme.

**Figure 3 entropy-23-01364-f003:**
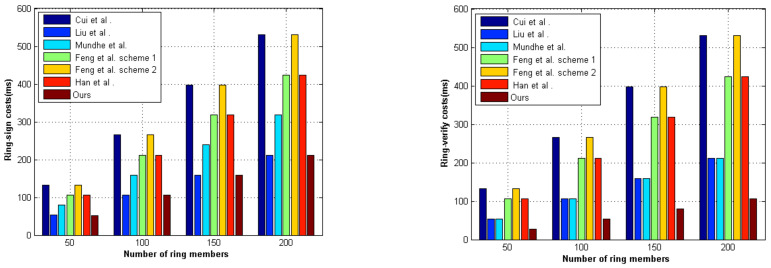
Computation costs of the different schemes.

**Figure 4 entropy-23-01364-f004:**
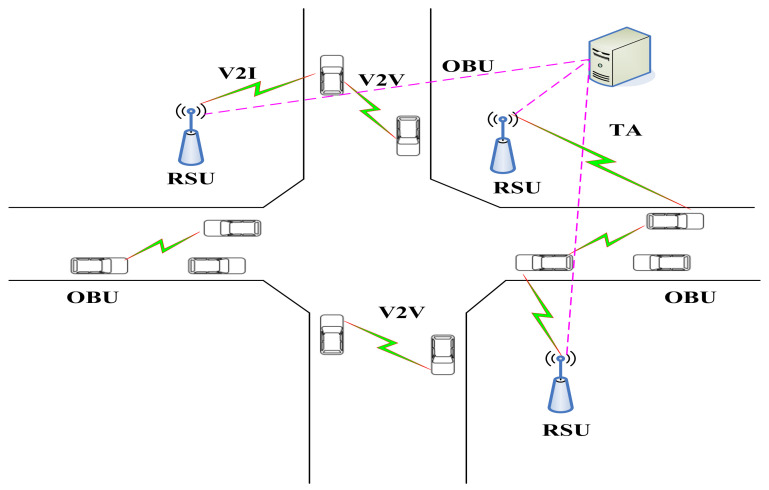
The architecture of a VANET.

**Figure 5 entropy-23-01364-f005:**
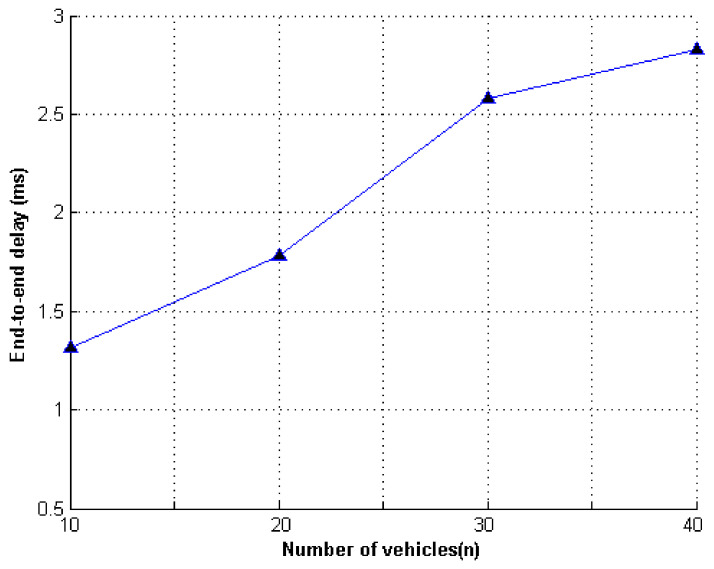
The vehicle density and E2ED delay.

**Figure 6 entropy-23-01364-f006:**
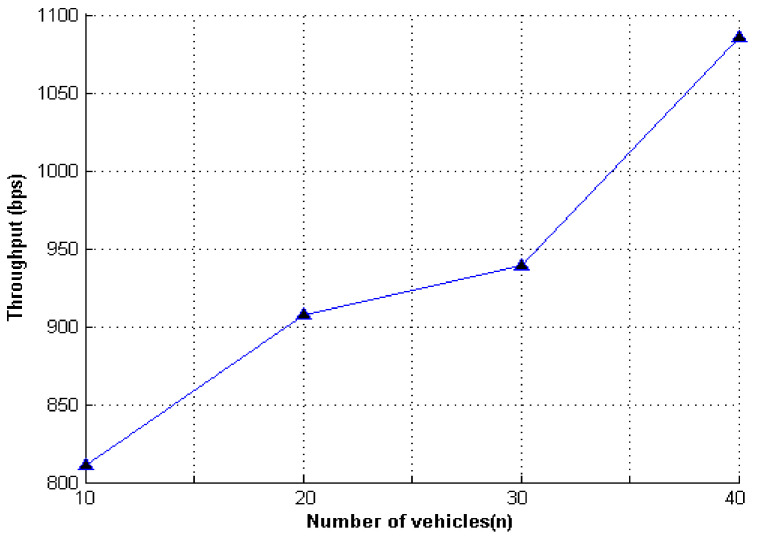
The vehicle density and data throughput.

**Table 1 entropy-23-01364-t001:** Parameter settings.

Parameter	Description	Sample
n	Polynomial ring degree	512
q	Large prime	$2^{25}$
m	Polynomial ring size	6
λ	Security parameter	100
δ	Hermite factor	1.007
κ	Random Oracle weight	14
η	Correctness	1.1
$\sigma_{1}=12\sqrt{\kappa }$	Rejection sampling	45
$M_{1}$	Rejection sampling	1.0027
$\sigma_{2}=12\eta \sigma_{1}\sqrt{m\kappa }$	Gaussian standard deviation	$2^{12.5}$
$M_{2}$	Rejection sampling	1.0027
$\sigma_{3}=12\eta \sigma_{2}\sqrt{m}$	Gaussian standard deviation	$2^{17.5}$
$M_{3}$	Rejection sampling	1.0027
Public key size	$n^{2}m\log2q$	4992 KB
Secret key size	$n^{2}m\log2q$	4992 KB
Signature size	nmlog(12σ)	672 KB

**Table 2 entropy-23-01364-t002:** Comparison of different security levels.

Security Level (bits)	Signature Size (KB)
100	672
128	684
256	768
512	825

**Table 3 entropy-23-01364-t003:** Comparison costs of relevant schemes.

Schemes	Ring-Sign Costs	Ring-Verify Costs	Signature Length
Cui et al. [16]	$5nT_{mult}{+T}_{h}$	$5nT_{mult}+T_{n}$	2(n+1)m
Liu et al. [17]	$2nT_{mult}+nT_{h}$	$2nT_{mult}+nT_{h}$	(n+1)m
Mundhe et al. [27]	$(3n+1)T_{mult}+2T_{h}$	$2nT_{mult}+2T_{h}$	(n+1)m
Feng et al. [21] scheme 1	$nT_{mult}+T_{n}+2T_{h}$	$nT_{mult}+T_{n}+2T_{h}$	(n+1)m
Feng et al. [21] scheme 2	$3nT_{mult}+T_{n}+2T_{h}$	$3nT_{mult}+T_{n}+2T_{h}$	(n+1)m
Han et al. [28]	$4nT_{mult}+2T_{h}$	$4nT_{mult}+2T_{h}$	(n+1)m
Ours	$(2n-1)T_{mult}+2T_{h}$	$(n+1)T_{mult}+2T_{h}$	nm+κ(κ≤m)

**Table 4 entropy-23-01364-t004:** Functionality comparison of relevant schemes.

Schemes	Unconditional Anonymity	Strong Unforgeability	Message Integrity
Wang et al. [11]	No	Yes	Yes
Cui et al. [16]	No	No	Yes
Liu et al. [17]	No	No	Yes
Mundhe et al. [27]	Yes	No	Yes
Feng et al. [21]	No	No	Yes
Han et al. [28]	Yes	Yes	Yes
Cai et al. [29]	No	No	Yes
Ours	Yes	Yes	Yes

**Table 5 entropy-23-01364-t005:** Simulation parameters.

Parameter	Value
Speed of vehicle	20 km/h
Transmission range	100 m
Time interval	2 s
MAC type	IEEE 802.11p
Number of lanes	4

## Data Availability

Not applicable.

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
