# Peer review of "Anti-Quantum Lattice-Based Ring Signature Scheme and Applications in VANETs"

_entropy, 2021, doi:10.3390/e23101364_

Round 1
Reviewer 1 Report
The paper covers a very interesting topic and the results have a great potential to be used in practical solutions. The overall structure of the paper is correct and has all the elements in place including an application example of the proposed solution. The paper is one of many similar papers within the Xi'an University of Technology team, so a lot of the text is very similar because it describes similar assumptions, but the individual papers present fundamentally different solutions. For example, section 3.1 is very similar to parts of section 4.1 Syntax of ring signature of the paper https://www.mdpi.com/1099-4300/23/10/1303.
Unfortunately, the work especially in the area of proposing your own solution is very difficult to read. For example (line 253) we can read: "performs the algorithm", what algorithm ? Alghorithm 1 or maybe 2?
The descriptions of the algorithms themselves are also in some places imprecise e.g. Algorithm 2 Ring-signing algorithm. What does the phrase "Continue with probability" mean - what exactly is the decision condition?
I have only given two examples which may lead to making the reception of the article extremely difficult. What I miss: a more detailed description of the proposed solution perhaps with examples, broader descriptions and above all drawings especially in the area of chapter 3 which is crucial. Many researchers will want to implement the proposed set of algorithms to be able to reproduce the results obtained and compare it with their own solution
At this stage, I make no comments on the sample implementation of the proposed article presented in Chapter 8. It is a superficially presented implementation, but in terms of the article presented it seems to be sufficient. I encourage the authors to discuss the results presented in Chapter 8 a bit more - perhaps in a separate article
In conclusion, I believe that the paper has the correct structure, but the way in which the proposed solution has been presented is very difficult to read. The content presented in the introduction based on a ready-made scheme, which can be found in many articles of the team with can remain unchanged. However, the description of the proposed solution needs to be expanded. Currently, in many places (I gave only two examples) it is difficult to unambiguously assess the proposed solution. If the article was published in this form, its readers would be limited only to a very narrow group of specialists working on a given solution.
Author Response
Dear reviewer,
Thank you very much for your valuable comments. Your comments are very helpful to improve the quality of this paper. In addition, we also asked MDPI to help with professional English editing. Please refer to the attachment。
Best regards
Chunhong Jiao

Reviewer 2 Report
Dear Authors,
The content of the article fits into the scope of an Entropy journal. One of the reasons is that the key research topic involves security enhancement issues based on the Authors' proposed an anti-quantum ring signature scheme based on lattice. It is very important in view of the unconditional anonymity and unforgeability.
The advantage is that this scheme is suitable for implementation in vehicular Ad-Hoc networks. It is relevant and interesting.
The above objective was based on more than 20 publications analysed in the initial sections of the article.
The results of the relevant simulation experiments in terms of end-to-end delay and throughput are given and explained. The manuscript contains some new data.
The “Conclusions” are consistent with the evidence and arguments presented and address to the main question posed.
NOTE: The article is presented in logical way but:
- Did the authors consider the choice of other parameters included in Table 1? To what extent is this set of parameters optimal and exhaustive?
- What was the criterion for justifying the choice of an average vehicle speed of 20km/h? Do the Authors foresee simulations for higher average vehicle speeds in the future?
- needs editorial improvement (e.g. i) references to literature sources in Tables 3 and 4 should be changed to appropriate ones; ii) the same should also be checked elsewhere in the text of the article); iii) remove "13." before Authors in ref. 14).
Author Response

(The authors gave the same response as above.)

Round 2
Reviewer 1 Report
The authors have addressed my comments and I am satisfied with the improvements. I believe that the article can be published in its present form.